# A Case of Curative Treatment with Apatinib and Camrelizumab Following Liver Resection for Advanced Hepatocellular Carcinoma

**DOI:** 10.3390/ijms241713486

**Published:** 2023-08-30

**Authors:** Julu Huang, Rong Liang, Cheng Lu, Lu Lu, Shuanghang Li, Minchao Tang, Xi Huang, Shilin Huang, Rongyun Mai, Xing Gao, Shizhuo Li, Can Zeng, Yan Lin, Jiazhou Ye

**Affiliations:** 1Department of Hepatobiliary Surgery, Guangxi Medical University Cancer Hospital, Nanning 530021, China; huangjulu@stu.gxmu.edu.cn (J.H.); lucheng@stu.gxmu.edu.cn (C.L.); tangminchao@stu.gxmu.edu.cn (M.T.); mairongyun@sr.gxmu.edu.cn (R.M.); li-sz2022@sr.gxmu.edu.cn (S.L.); zc0511004179@sr.gxmu.edu.cn (C.Z.); 2Department of Digestive Oncology, Guangxi Medical University Cancer Hospital, Nanning 530021, China; liangrong@gxmu.edu.cn (R.L.); lulu@stu.gxmu.edu.cn (L.L.); lishuanghang@stu.gxmu.edu.cn (S.L.); huangxi@stu.gxmu.edu.cn (X.H.); huangshilin@stu.gxmu.edu.cn (S.H.); gaoxing@stu.gxmu.edu.cn (X.G.)

**Keywords:** advanced hepatocellular carcinoma, tyrosine kinase inhibition, PD-1 inhibitor, curative liver resection, HCC

## Abstract

Hepatocellular carcinoma (HCC), a highly malignant digestive system tumor, poses substantial challenges due to its intricate underlying causes and pronounced post-surgery recurrence. Consequently, the prognosis for HCC remains notably unfavorable. The endorsement of sorafenib and PD-L1 inhibitors for HCC signifies the onset of a new era embracing immunotherapy and targeted treatment approaches for this condition. Hence, comprehending the mechanisms underpinning targeted immune combination therapy has become exceedingly vital for the prospective management of HCC patients. This article initially presents a triumphant instance of curative treatment involving the combination of TKI and PD-1 inhibitor subsequent to liver resection, targeting an advanced stage HCC as classified by the BCLC staging system. The case patient carries a decade-long history of hepatitis B, having undergone a regimen of 20 courses of treatments involving apatinib and camrelizumab. Throughout the treatment period, no occurrences of grade 3 or 4 adverse events (AE) were noted. Subsequently, the patient underwent a left hepatectomy. Following the hepatectomy, their serum AFP levels have consistently remained within normal limits, and CT imaging has indicated the absence of tumor recurrence over a span of 36 months. The patient had been reviewed on time for two years after the operation. The last time a CT was performed for this patient in our hospital was 7 May 2021, and no new tumors were found. Follow-up is still ongoing. When applying combined targeted immune transformation therapy using TKI and ICI for a patient with BCLC advanced stage HCC, apatinib treatment serves a dual purpose. It inhibits the survival and angiogenesis of tumor cells, while also enhancing the efficacy of camrelizumab in obstructing the interaction between PD-1 and PD-L1. This restoration of T cell cytotoxicity subsequently facilitates the elimination of tumor cells, leading to an enhanced anticancer effect.

## 1. Introduction

HCC ranks as the sixth most prevalent cancer globally and stands as the third primary contributor to cancer-related fatalities [1]. The prognosis for hepatocellular carcinoma remains notably grim due to challenges in early detection and a high post-operative recurrence rate. However, recent advancements in targeted therapy, immunotherapy, surgical techniques, and localized treatments have ignited fresh optimism for individuals grappling with advanced liver cancer. The FDA sanctioned sorafenib’s usage for addressing unresectable HCC [2]. This pivotal approval heralded the commencement of the targeted therapy era for hepatocellular carcinoma. Sorafenib, a multi-kinase inhibitor, exhibits dual capabilities in inhibiting tumor growth. It effectively targets various cell surface tyrosine kinases including vascular endothelial growth factor receptor 1 (VEGFR-1), vascular endothelial growth factor receptor 2 (VEGFR-2), platelet-derived growth factor receptors-β (PDGFR-β), and BRAF, among others [3,4]. Within HCC, sorafenib exerts an anticancer effect by impeding the proliferation of hepatocellular carcinoma cell lines and prompting apoptosis. Researchers have observed that the transition from hepatitis to liver cirrhosis to liver cancer is accompanied by a gradual weakening of immune cell functions in the body. This phenomenon facilitates tumor immune escape and consequently amplifies the invasive potential of the tumor. Consequently, systemic treatment is often advocated for in advanced liver cancer cases. Currently, prominent approaches encompass targeted therapy, immune checkpoint inhibitors (ICIs), combined therapies, and systemic chemotherapy. Notably, the realm of systemic HCC treatment has witnessed noteworthy advancements in recent years. This includes the introduction of agents like sorafenib, lenvatinib, regorafenib, cabozantinib, ramucirumab, as well as the emergence of immune-based options such as nivolumab and pembrolizumab. These systemic medications have gained FDA approval for addressing advanced hepatocellular carcinoma [5]. The first-line targeted drugs are mainly sorafenib and lenvatinib [6], both of which are tyrosine kinase inhibitors. Sorafenib effectively curbs HCC cell proliferation by impeding the RAF/MEK/ERK signaling pathway. Furthermore, it hinders angiogenesis by targeting VEGFR and platelet-derived growth factor receptors (PDGFRs). This dual action contributes to apoptosis induction and underscores its anticancer effectiveness [7]. Lenvatinib’s anti-angiogenic effects primarily stem from its ability to inhibit VEGF and FGF signaling pathways [8]. Moreover, various international multi-center clinical trials have underscored apatinib’s potential as an anti-HCC agent, manifesting a prolonged overall survival for patients. Encouragingly, in the majority of instances, the management of toxicities and side effects has proven feasible [7,9,10,11,12,13]. As a result, apatinib has garnered consideration as a potential second-line targeted therapy for advanced HCC. Apatinib’s anti-tumor mechanism revolves around its capacity to modulate the phosphorylation levels of PDGFR-α, insulin-like growth factor 1 receptor (IGF-IR), and AKT via VEGFR inhibition. This disrupts the PI3K/AKT signaling pathway inhibition, culminating in cell cycle arrest within the G2/M phase and instigating apoptosis in HCC cells. Furthermore, some researchers have illuminated apatinib’s potential in countering drug resistance mechanisms in HCC, further highlighting its multifaceted therapeutic potential [7].

The current immunotherapy drugs, ICIs, include the following: anti-PD-1 (pembrolizumab and nivolumab), anti-PD-L1 (durvalumab and atezolizumab), and anti-CTLA-4 (tremelimumab and ipilimumab) [14]. They have been shown to be effective in the treatment of patients with hepatocellular carcinoma. PD-1 can produce inhibitory signals by binding to PD-L1, thereby inhibiting the activation of these immune cells and protecting tumor cells from attack [15]. Anti-CTLA-4 antibody can enhance the anti-tumor effect of T cells by blocking the binding of CTLA-4 on T cells to CD80/86 on APC [14]. However, the combination therapy of TKI and ICIs has been proven to be more effective than single targeted or single immunotherapy in many tumors. So, an in-depth study of the mechanism of TKI combined with ICIs brings new hope regarding the treatment of other cancer species, including advanced liver cancer.

Some of the possible predictive biomarkers under study can guide individualized treatment strategies and improve the prognosis of HCC patients. For example, the expression levels of angiogenesis biomarkers, VEGF and other angiogenesis-related markers, can predict the efficacy of anti-angiogenesis drugs (such as sorafenib and rivantinib) [16]. *TERT, CTNNB1, TP53, FGF19,* and *TP53* were found to be predictors of ICI efficacy [14]. Ma et al. found that *RAD54 B* was selected as an independent risk factor for the prognosis of patients with *LNM* of liver cancer using a regression model, and its expression was significantly positively correlated with tumor mutation load and microsatellite instability in high-risk subtypes [17]. Regarding α-fetoprotein (AFP), baseline AFP levels were associated with response and survival outcomes in patients receiving systemic therapy [18].

## 2. Case Presentation

A 39-year-old man sought care at our hospital clinic, reporting persistent upper abdominal pain spanning two weeks. Notably, he carries a history of hepatitis B spanning over a decade, during which he has not undergone anti-HBV treatment. Laboratory tests showed that his serum AFP level was 120,000 ng/mL. Simultaneously, contrast-enhanced abdominal CT scans unveiled diffuse mass lesions displaying typical liver malignancy features within segments II, III, IVa, IVb, V, and VIII. Notably, a tumor thrombus had infiltrated the left hepatic vein (LHV), middle hepatic vein (MHV), the main trunk of the portal vein (MPV), and the left branch of the portal vein (LPV). Encouragingly, no indications of extrahepatic metastasis were observed. Positron emission tomography–computed tomography (PET–CT) scans highlighted the localized abnormal accumulation of fluorodeoxyglucose (FDG) within the tumor, exhibiting a maximum standardized uptake value (SUVmax) of 8.4, arousing a suspicion of malignancy. The subsequent histological analysis of a liver tumor biopsy specimen confirmed the presence of hepatocellular carcinoma (HCC). In alignment with these findings, a diagnosis of advanced stage hepatocellular carcinoma according to the Barcelona Clinical Liver Cancer (BCLC) classification was established. 

The patient’s Child–Pugh liver function evaluation yielded a score of five, corresponding to grade A. Consequently, the patient underwent twenty cycles of treatment involving apatinib (a multi-Tyrosine kinase inhibitor) at an oral dose of 150 mg/day, along with camrelizumab (a PD-1 inhibitor) administered intravenously at a dose of 200 mg every 15 days. Additionally, a routine daily oral dose of 0.5 mg entecavir was administered for anti-HBV therapy. Only four months later, the serum AFP level dramatically decreased and stayed within the normal limits, as the serum AFP level was 7.3 ng/mL, and the contrast-enhanced abdominal CT images revealed that the primary liver tumor markedly shrunk. By the tenth month following treatments, only a few diffuse mass lesions persisted in the left hemiliver, while the tumor thrombi in the left hepatic vein (LHV), middle hepatic vein (MHV), main trunk of the portal vein (MPV) had completely vanished. PET–CT scans showcased reduced FDG accumulation, with a lowered SUVmax of 4.5 in the remaining tumor. Consequently, the apatinib and camrelizumab treatments yielded a partial response (PR) therapeutic effect according to the modified Recist (mRecist) criteria for this patient. Importantly, no grade 3/4 adverse events (AE) occurred during the course of treatment. Given the presence of remnant tumors confined to the left hemiliver, along with a favorable general condition boasting a PS score of 0 and Child–Pugh A liver function, the patient proceeded to undergo a left hepatectomy. Since the liver resection, the serum AFP levels have stayed within the normal limits, i.e., the serum AFP levels have fluctuated between 2.34 ng/mL and 1.83 ng/mL, and the CT images have not detected any tumor recurrence over the course of 36 months. This represents a successful case wherein curative therapy involving TKI in conjunction with a PD-1 inhibitor was pursued subsequent to liver resection for an advanced stage BCLC HCC. The patient’s complete medication journey is outlined in Figure 1.

The patient’s pre-treatment CT scan unveiled the presence of a massive hepatocellular carcinoma spanning the left inner lobe and right anterior lobe of the liver (measuring 11.8 × 8.3 cm). This carcinoma was accompanied by a tumor thrombus that had infiltrated the left hepatic vein (LHV), middle hepatic vein (MHV), main trunk of the portal vein (MPV), and the left branch of the portal vein (LPV). Concurrently, PET–CT images exhibited a focal and abnormal accumulation of fluorodeoxyglucose (FDG) within the tumor, reflecting a SUVmax value of 8.4, thereby raising the suspicion of malignancy. At the same time, the serum AFP level was 109,464 ng/mL. However, the first AFP concentration of 120,000 ng/mL was detected outside the hospital. Consequently, a targeted immune combination therapy was administered from 7 December 2018, to 26 March 2019. Subsequently, following 3, 7, and 8 months of the targeted immune combination treatment, in conjunction with the findings from imaging assessments, the mRecist criteria were used to evaluate the patient’s response as a partial response (PR). On 7 April 2019, the patient underwent an extended left hemihepatectomy. Subsequent follow-ups at 1 month, 3 months, 6 months, 1 year, and 2 years, as determined through CT scans, revealed stable disease (SD). Throughout this period, the serum AFP level exhibited a range between 2.34 ng/mL and 7.21 ng/mL. (Abbreviations: AFP—alpha fetoprotein; TMB—tumor mutation burden; CT—computed tomography; HCC—hepatocellular carcinoma; PET–CT—positron emission tomography–computed tomography).

Why did apatinib alongside camrelizumab treatment contribute to a promising therapeutic effect for advanced HCC?

Although the BCLC treatment algorithm recommends sorafenib or lenvatinib as the preferred first-line treatment for advanced HCC, both the SHARPE and Oriental trials [19] demonstrated that the limited therapeutic effect of sorafenib meant it only prolonged the overall survival (OS) period by approximately 3 months compared with placebo. The REFLECT trial [20] revealed that lenvatinib did not provide a better survival benefit regarding OS and progression-free survival (PFS) compared with sorafenib.

Consequently, utilizing a single TKI as the initial treatment for advanced HCC often falls short of satisfactory results. In contrast, a more comprehensive systemic anticancer approach combining apatinib and camrelizumab was chosen for this patient. Supplementary to monitoring serum AFP levels, continuous assessments of serum tumor mutation burden (TMB) and cfDNA concentration served as additional tools for prognostic surveillance. However, these markers did not consistently align with changes in serum AFP levels or imaging characteristics.

Interestingly, the HCC sample exhibited a negative PD-L1 expression. Given these complexities, the compelling efficacy of the apatinib and camrelizumab regimen in achieving a remarkable anticancer effect for this patient warrants further investigation. The combination’s ability to generate such positive outcomes, despite the unconventional markers, emphasizes the need for a deeper understanding of its underlying mechanisms.

To elucidate the underlying mechanism driving the notable therapeutic outcomes observed in this patient, we undertook next-generation sequencing (NGS) and subsequently conducted rigorous bioinformatics analysis. Performing Kyoto Encyclopedia of Genes and Genomes (KEGG) enrichment analysis on the whole-exome sequencing (WES) data extracted from the HCC tissue before treatment, we identified a notable aggregation of high-frequency mutation genes within signaling pathways that foster tumorigenesis and modulate drug metabolism (Figure 2A–C, Table 1). Notably, the MAPK pathway exhibited robust activation, while the PI3K-Akt pathway and apoptosis regulation were prominently suppressed. Upon achieving a significant partial response (PR) therapeutic effect in this patient, we proceeded with the RNA sequencing of the HCC tissue. This was followed by the construction of a visual analysis aimed at comparing the SNP and CNV events in this patient with those in a group of TCGA HCC patients who had not received any medication (Figure 2D,E). Subsequently, we identified a control cohort comprising 58 TCGA HCC patients with similar SNP and CNV event profiles to this patient, as depicted in Figure 2F. Through KEGG functional enrichment analysis, it was evident that post-treatment, there was a significant inhibition of pathways including VEGFA/VEGFR and MAPK, while PD-L1 exhibited an up-regulation (Figure 3A). Furthermore, an increase in the abundance of T cells and NK cells was observed, coupled with a notable decrease in regulatory T cells (Tregs) (Figure 3B), in comparison with the corresponding metrics in the TCGA cohort.

In alignment with these findings, the mechanisms underlying the remarkable therapeutic outcomes observed in this patient have been elucidated. Illustrated in Figure 4, apatinib exerted a profound reduction in VEGFA/VEGFR transcription, directly targeting the downstream HRAS–MAPK3–FOS signaling cascade. This dual action effectively curbed cell proliferation and hindered DNA damage repair, whether in vascular endothelial cells or tumor cells. This combined effect translated to the suppression of angiogenesis and cellular survival within the tumor microenvironment. Of note, an intriguing feedback mechanism comes into play within tumor cells, where the HRAS–MAPK–FOS cascade reaction generates a subsequent reduction in the production and release of VEGFA. This intricate interplay further reinforces the antiangiogenic and cell survival functions within the tumor milieu. Simultaneously, *HRAS* emerges as a pivotal target for orchestrating various anticancer effects within tumor cells. The inhibition of HRAS directly triggered the activation of *PIK3CA*, subsequently inducing apoptosis in tumor cells. Notably, while *PIK3CA* promoted PD-L1 expression to potentially aid in immune escape, the administration of camrelizumab disrupted the PD-L1 interaction with the T cell surface receptor PD-1, thus obstructing the apoptotic signal in T cells. In summary, this case offers insights into the intricate interplay between the immune and tumor microenvironments. The combined effect of apatinib treatment encompassing the inhibition of tumor cell survival and angiogenesis, coupled with camrelizumab’s capacity to hinder PD-1 binding to PD-L1, contributes to the restoration of T cell cytotoxicity against tumor cells. This orchestrated approach enhances the overall anticancer effect, as depicted in Figure 4.

To date, the combination of TKIs with immune checkpoint inhibitors has brought significant diversification to systemic anticancer treatment approaches. Notably, in the United States, over 75% of oncologists have incorporated NGS tests to guide treatment decisions, even for patients with advanced refractory diseases, starting from 2017. However, limited studies have delved into the intricate influence of bioinformatics on both the tumor and immune microenvironments regarding the anticancer effect. This case study furnishes valuable evidence and a reference point for devising strategies that employ NGS tests to select the most suitable anticancer regimen and effectively monitor the prognosis of advanced HCC treatment.

## 3. Discussion

In order to confirm that the case patient had a genetic profile typical of liver cancer patients, we compared the CNVs and SNPs of the case patient with typical liver cancer patients in the TCGA database. Our hypothesis was that tumor sensitivity was mediated not by a single genetic factor but by the accumulation of genetic alterations throughout the genome. Therefore, we explored the biological functions and signaling pathways involving all mutated genes that might play a role in carcinogenesis. Our goal was not only to clarify how patients may show a strong sensitivity to the combination therapy of apatinib and camrelizumab, which would also reveal the mechanism of action of this combination therapy, but also to identify potential biomarkers that might predict the efficacy of such therapy. As a control group, we selected 58 patients with clinically similar liver cancer who were not treated with the combination of apatinib and camrelizumab. We first identified differentially expressed genes between patients treated or not with combination therapy, and then we examined the functions and pathways enriched in those differentially expressed genes. We also explored differences in the profile of immune cells infiltrating tumor tissue between the case patient on combination therapy and LIHC patients in the TCGA database. We subjected the gene sets potentially affected by combination therapy to gene set variation analysis, generating a treatment score for each gene. We explored the potential correlation of this score with patient survival and prognosis.

Our patients who responded well to the combination of apatinib and camrelizumab had harmful mutations in the tumor, which affected the genes *ZNF717, CDC27, TP53, KCNJ12, KCNJ18,* and *PDE4DIP*. The observed TP53 mutations reduce the anti-tumor ability of TP53, enabling rapid tumor development [21]. *ZNF717* participates in cell proliferation, differentiation, and apoptosis, and mutations in the gene may contribute to colorectal [22] and liver cancer [23]. *CDC27* may promote the proliferation [24] and metastasis [25] of colorectal cancer cells. *KCNJ12* may contribute to the initiation and development of carotid paraganglioma, while mutations in *KCNJ18* increase the risk of pancreatic cancer [26]; mutations in *KCNJ12* and *KCNJ18* might contribute to multiple types of cancer [27]. *PDE4DIP* mutations may contribute to pinealoblastoma.

However, these mutations were not the specific mutations of cancer. Most of the 263 HCC patients in the cohort had a mutation in at least one of the genes discussed above, except for mutations in *KCNJ18, ZNF717*, or *RBMX*. We concluded that our case patient showed the classical genomic profile of hepatoma. In addition, the mutant gene of the case patient was involved in PI3K-Akt signaling, *MAPK* signaling, apoptosis, and drug metabolism. Abnormal gene expression in these pathways, or in the genes that act upstream to regulate these pathways, might promote tumor occurrence.

Combination therapy inhibited *MAPK* signaling, *PI3K-Akt* signaling, and immunosuppression by tumor-associated Tregs. At the same time, it up-regulated PD-L1 expression, activated the PD-1 checkpoint pathway, and increased the numbers of cytotoxic immune cells such as T cells and NK cells. These results might help to explain the observed efficacy of the combination of camrelizumab and apatinib against liver cancer. Altogether, our results indicated that the case patient showed differences to the control patient cohort in their transcriptome and in the profile of tumor infiltration by immune cells.

Our results suggest that patients showing a lower treatment score than the case patient, based on a gene set variation analysis of potential mechanistic genes, might be more likely to experience clinical benefit from combination therapy. Our results further suggest that the following genes may be biomarkers of drug efficacy in those HCC patients who were negative for PD-L1 expression or who were resistant to TMB: *PDCD1, MMP9, RPS6, NFKBIB, PPP2R1A, HRAS, CDC37, MLST8, MAPK3, FOS, CD274, VEGFA, RHEB, RELA, ATF6B, CRTC2, TICAM2, CSNK2A3, BRAD, PIK3CA, TRAF6, PTEN,* and *MYD88*. After calculating the treatment score for 424 samples, we found that 352 (83%) were predicted to respond well to combination therapy, including patients with a similar genomic profile. Surprisingly, the treatment score appears to predict not only the efficacy of combination therapy but also the survival and prognosis of patients. This might make the score quite useful in the clinic.

## 4. Conclusions

Our results supported an innovative systemic treatment combining camrelizumab with apatinib against HCC. By up-regulating PD-L1 expression, activating the PD-1 checkpoint pathway, and inhibiting MAPK and PI3K-Akt signaling, combination therapy might effectively inhibit immunosuppression by tumor-associated Tregs while increasing the numbers and activity of killer immune cells such as T cells and NK cells. Our findings justify conducting further basic and clinical studies on this combination therapy. However, the combination of TKI and immunotherapy in general is not always the best option. There are studies of combination treatments with negative results. But, we think that the specific combination that we used seems to be beneficial.

A key insight from our analysis is that combination therapy appears to involve interactions between MAPK and PI3K-Akt signaling pathways. By inhibiting these pathways, the therapy inhibits tumor cell proliferation and angiogenesis. An overexpression of *CD274* in tumor cells may lead more of them to be bound with the *PDCD1* of T cells. The combination therapy appears to up-regulate PD-L1 and activate the PD-1 checkpoint pathway, increasing the abundance of cytotoxic immune cells such as T cells and NK cells while inhibiting immunosuppression by tumor-associated Tregs. This may help to explain the observed efficacy of the combination of camrelizumab and apatinib against HCC.

## Figures and Tables

**Figure 1 ijms-24-13486-f001:**
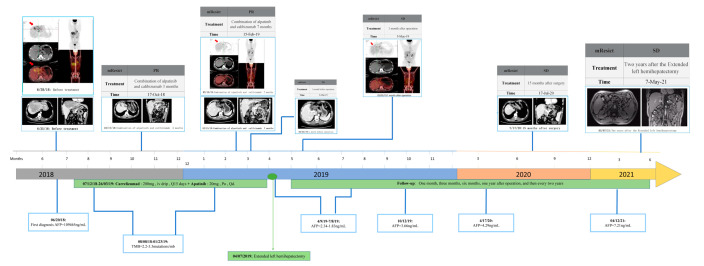
Summary of medication history.

**Figure 2 ijms-24-13486-f002:**
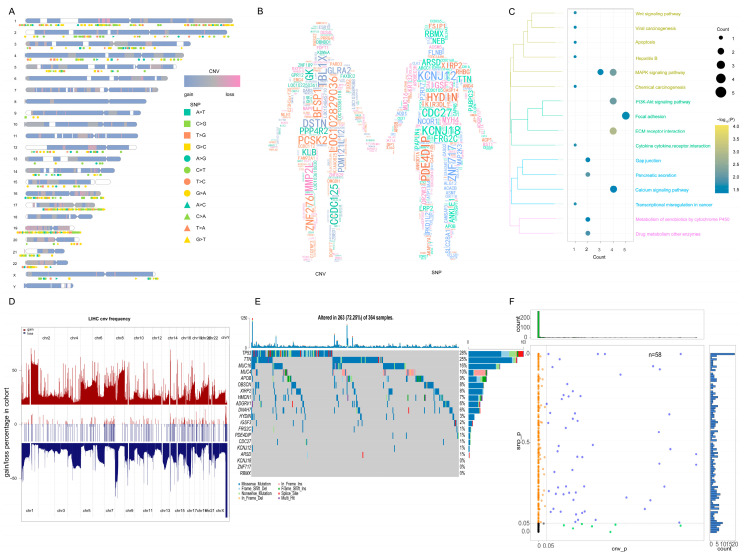
Genomic information for the case patient. (**A**) CNV and SNP displayed on chromosome. The chromosome numbers are listed on the left. The different shapes and colors of SNP represent different mutation types. The two different colors of CNV represent the gain or loss. (**B**) CNV and SNP of patients in human body. Gene font was directly proportional to CNV and SNP of HCC case patient, and gene color was random. (**C**) CNV and SNP genes were significantly involved in the pathway. In KEGG pathway, the bubble size in the pathway is the number of genes involved in the pathway, and the bubble color is the enrichment significance. (**D**) The case patient had genomic similarity with TCGA LIHC patients. The middle region is the CNV of HCC case patient, and the upper and lower regions are the CNV of TCGA patients with liver cancer, indicating that CNV is also a common CNV. (**E**) The high frequency mutation gene of case patient in the TCGA LIHC patient cohort. The top 20 genes with mutation frequency were displayed in 364 samples; additionally, 263 samples had genes with high frequency mutation, indicating that the HCC case patient SNP is also a common SNP. (**F**) The fifty-eight samples with *p* value greater than 0.05 were obtained via a chi-square test. The 58 samples were genomically similar to the HCC case patient. CNV, copy number variation; SNP, single nucleotide polymorphism; HCC, hepatocellular carcinoma; TCGA, the cancer genome atlas; LIHC, liver hepatocellular carcinoma.

**Figure 3 ijms-24-13486-f003:**
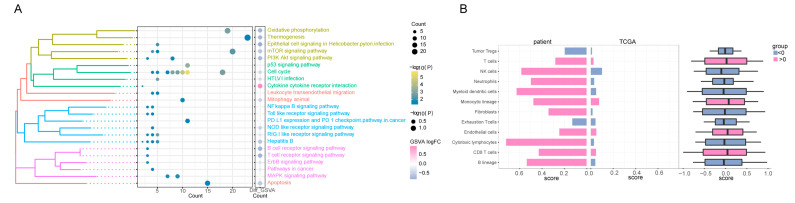
Exploring the KEGG signal pathway and immune infiltration affected by combined therapy. (**A**) The KEGG pathway with the significant participation of differentially expressed mRNA. The bubble size color in the bubble chart on the left represents the gene count and in a specific pathway. The right side of the bubble shows the score of the pathway gene, and its color, high, and low expression represent the activation and inhibition state of the pathway. (**B**) The difference in immune infiltration between the HCC case patient and TCGA LIHC clinical cohort represent genomic similarity and genome difference. The left side of the figure is the immune infiltration score of the patient after treatment; the right side is the immune infiltration score of 58 patients similar to the patients in TCGA. Red represents a positive and blue represents a negative; the higher the scores, the more cell infiltrates in the patient. KEGG: Kyoto Encyclopedia of Genes and Genomes; HCC: hepatocellular carcinoma; TCGA: The Cancer Genome Atlas; LIHC: liver hepatocellular carcinoma.

**Figure 4 ijms-24-13486-f004:**
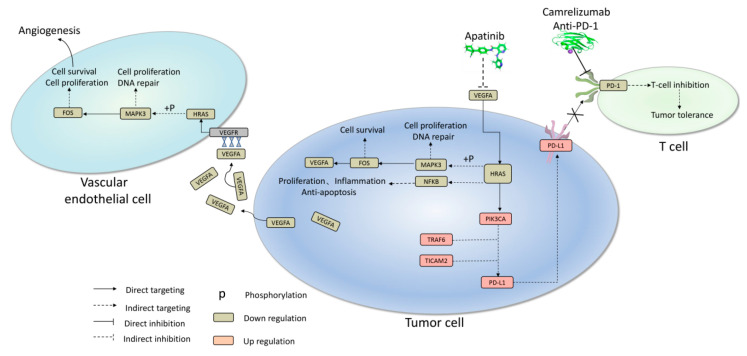
Major mechanism underlying apatinib combined with camrelizumab achieved a promising anti-cancer effect in this case. X: Blocking the pathway prevents PD-1 from binding to PD-L1.

**Table 1 ijms-24-13486-t001:** Genes related to cancer and drug metabolism pathways involved in the CNV and SNP of patients.

**Before Treatment**				
**Pathway Name**	***p* Value**	**Fdr**	**Gene Name**	**Gene Status**
MAPK signaling pathway	0.006894022	1	FLNB/TP53/HSPA6MAP2K3	activation
PI3K-Akt signaling pathway	0.008034318	1	COL24A1/COL6A6/COL4A3/COL6A2	inhibition
Apoptosis	0.033360695	1	ATM/BCL2/TP53	inhibition
Hepatitis B	0.044232577	1	CDK2/CDKN1B/CDKN1A/TP53	activation
**After Treatment**				
**Pathway Name**	***p* Value**	**Fdr**	**Gene Name**	**Gene Status**
MAPK signaling pathway	0.003797448	0.318171929	BRAF/RAP1B/MAPK3/NF1/MAP3K1/HRAS/RASA2/	inhibition
PI3K-Akt signaling pathway	0.019029251	0.598612574	PTEN/CRTC2/CDC37/MLST8/ATF6B/RHEB/RPS6/PPP2R1A	activation
Apoptosis	0.045364706	0.476518519	ACTG1/ATM/BIRC5/DAXX/FOS/GADD45B/HRAS/IL3RA/LMNA/MAPK3/PIDD1/PIK3CA/RELA/TRADD/TUBA3C	activation
Hepatitis B	0.047111607	0.774766417	MAPK3/ATF6B/RELA/FOS	inhibition
PD-L1 expression and PD-1 checkpoint pathway in cancer	0.040184742	0.476518519	CSNK2A3/FOS/HRAS/MAPK3/MYD88/NFKBIB/PIK3CA/PTEN/RELA/TICAM2/TRAF6	activation
NF-kappa B signaling pathway	0.01547192	0.541748109	TRAF6/TICAM2/MYD88/TRADD	inhibition
T cell receptor signaling pathway	0.039225403	0.701472906	MAPK3/FOS/HRAS	activation
p53 signaling pathway	0.010435202	0.200648803	ATM/ATR/CCNB1/CCNB2/CDK1/GADD45B/GTSE1/PIDD1/PTEN/SFN/SHISA5	activation

## Data Availability

The original contributions presented in the study are included in the article/Appendix A; further inquiries can be directed to the corresponding author/s.

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
