# Peer review of "A Case of Curative Treatment with Apatinib and Camrelizumab Following Liver Resection for Advanced Hepatocellular Carcinoma"

_ijms, 2023, doi:10.3390/ijms241713486_

Round 1

Reviewer 1 Report

The submitted manuscript entitled “A case of curative treatment of apatinib associated calibizumab following liver resection for advanced hepatocellular carcinoma” focuses on the case study of efficacy of systemic treatment with use of apatinib combined with calibizumab and liver resection of a 39-year man. The authors also attempted to study mechanisms underlying anticancer effects of used drugs. This study scientifically sounds and may be of interest for the journal audience. The manuscript contains 18 references and 10 of them were published last 5 year. 4 Figures and 1 Table are presented to illustrate the results obtained. However, there are some concerns and recommendations to improve the quality of the manuscript. They are as follows:

1.     Overall, the manuscript is inaccurately written. There are many mistakes in English grammar and style and lack of gaps etc. Especially, this concerns the Abstract.

2.     In the Abstract section, line 29: “The patient is still alive nowadays”. This should be clarified - for how long time. Then, if take into account that Figure 1 depicts the patient history only till the mid of 2021.

3.     On lines 49-59, the authors attempted to discuss shortly immune system functions and immunotherapy approaches. However, this was done very superficially and simplified; this text does not sound as a scientific one. This fragment of the Introduction section has to be significantly revised. It is recommended to discuss in more details systemic therapy of HCC patients and usefulness of predictive biomarkers; the authors can use the following papers: doi: 10.1080/14737159.2021.1987217, doi: 10.4254/wjh.v12.i10.766, doi: 10.1182/bloodadvances.2020001834. 

4.     In the Case presentation section, Figures 1,2,3 are very small and with low resolution. Figure captions are poorly readable.

5.     On line 82 and line 92, it is stated that AFP level stayed within normal limits. What AFP levels the authors mean? Also, the authors stated on line 66 that before treatment serum AFP level was 120,000 ng/ml. In the same time, on line 106, they stated that this level was 109,464 ng/ml/

6.     It is recommended to give short description of methods (NGS, bioinformatic analysis, KEGG enrichment analysis) used in this study.

7.     The references 5, 7 and 10 are incomplete. The reference 16 contains inaccurate information: Molecular Biology should be changed for Mol Biol (Mosk).    

Extensive editing is required

Reviewer 2 Report

It is a very well presented study. The results are amazing. 

i have to make some minor comments

1. in line 81 you declare that the patient received 20 mg entecavir against HBV. You must correct it. (the dose of entecavir is 0.5 mg per day)

2. figure 1 must be enlarged. It is very difficult to see the details of the figure

3. In discussion you must point that the combination of TKI and immunotherapy in general is not always the best option. There are studies of combination treatments with negative results. You must clarify that the specific combination that you used seems to be beneficial.   

There are many grammar and syntax problems. Please correct 

Author Response

1: in line 81 you declare that the patient received 20 mg entecavir against HBV. You must correct it. (the dose of entecavir is 0.5 mg per day). --- We modify as:We have modified it, Please check it again.

2: figure 1 must be enlarged. It is very difficult to see the details of the figure. --- We modify as:Because the resolution of the original picture is too high, I compressed it before uploading it to the submission system. Now I am sending the file of the original picture to “Resubmit Manuscript” in the submission system. We upload them as a compressed file.

3: In discussion you must point that the combination of TKI and immunotherapy in general is not always the best option. There are studies of combination treatments with negative results. You must clarify that the specific combination that you used seems to be beneficial. --- We modify as:We have added this paragraph on page 275 of the article: However,the combination of TKI and immunotherapy in general is not always the best option. There are studies of combination treatments with negative results. But we think that the specific combination that we used seems to be beneficial. Please check it again.

Round 2

Reviewer 1 Report

The authors have some revisions; however, some concerns remain non-addressed. As the authors’ response stated, “we have not modified this paragraph yet”. These concerns should be properly addressed before publication:

1.     Previous concern 1. The authors stated “We have decided to send the manuscript to a professional institution for touching up manuscript, which will take about 4-6 days”. I hope, that there has been enough time to edit the manuscript and this will be done properly.

2.     Previous concern 3. In the Introduction, sufficient discussion of efficacy and safety of immunotherapy during the perioperative period in hepatocellular carcinoma patients should be provided. Additionally, it is STRONGLY RECOMMENDED TO CHECK THE MANE OF A SECOND DRUG, ANTI-PD-1 ANTIBODY. It seems that calibizumab is not correct because I found no one paper in scientific literature with this name. I suggest that a combination therapy includes apatinib plus CAMRELIZUMAB, but not apatinib plus calibizumab. Please, check this. See and discuss the following papers: doi: 10.1136/jitc-2022-004656 and doi: 10.1158/1078-0432.CCR-20-2571. For predictive biomarkers in HCC see and cite the following paper: doi: 10.1080/14737159.2021.1987217. And for apatinib see and cite the following paper: doi: 10.4254/wjh.v12.i10.766. 

3.     Previous concern 5. As the authors stated, the two different baseline AFP level were given because the fist AFP concentration of 120,000 ng/ml was detected outside the hospital, while the second concentration od 109,464 ng/ml was detected in the hospital. Therefore, what the authors should do is to write this in the text of “Case presentation” section.

English language requires polishing
